# Impact of left ventricular assist devices and heart transplants on acute myocardial infarction and heart failure mortality and readmission measures

Eric J. Brandt[ID][1], Joseph S. Ross[ID][2,3,4], Jacqueline N. Grady[2], Tariq Ahmad[1,2], Sumeet Pawar[1], Susannah M. Bernheim[2], Nihar R. Desai[1,2]*

**1** Section of Cardiovascular Medicine, Department of Internal Medicine, Yale University School of Medicine, New Haven, CT, United States of America, **2** Center for Outcomes Research and Evaluation, Yale-New Haven Hospital, New Haven, CT, United States of America, **3** Section of General Medicine, Department of Internal Medicine, Yale University School of Medicine, New Haven, CT, United States of America, **4** Department of Health Policy and Management, Yale University School of Public Health, New Haven, CT, United States of America

* nihar.desai@yale.edu

**Data Availability Statement:** Because access to the data used in this study was made possible through both purchasing and data use agreements

## Abstract

### Background

Concern has been raised about consequences of including patients with left ventricular assist device (LVAD) or heart transplantation in readmission and mortality measures.

### Methods

We calculated unadjusted and hospital-specific 30-day risk-standardized mortality (RSMR) and readmission (RSRR) rates for all Medicare fee-for-service beneficiaries with a primary diagnosis of AMI or HF discharged between July 2010 and June 2013. Hospitals were compared before and after excluding LVAD and heart transplantation patients. LVAD indication was measured.

### Results

In the AMI mortality (n = 506,543) and readmission (n = 526,309) cohorts, 1,166 and 1,016 patients received an LVAD while 3 and 2 had a heart transplantation, respectively. In the HF mortality (n = 1,015,335) and readmission (n = 1,254,124) cohorts, 789 and 931 received an LVAD, while 212 and 202 received a heart transplantation, respectively. Less than 2% of hospitals had either ≥6 patients who received an LVAD or, independently, had ≥1 heart transplantation. The AMI mortality and readmission cohorts used 1.8% and 2.8% of LVADs for semi-permanent/permanent indications, versus 73.8% and 78.0% for HF patients, respectively. The rest were for temporary/external indications. In the AMI cohort, RSMR for hospitals without LVAD patients versus hospitals with ≥6 LVADs was 14.8% and 14.3%, and RSRR was 17.8% and 18.3%, respectively; the HF cohort RSMR was 11.9% and 9.7% and RSRR was 22.6% and 23.4%, respectively. In the AMI cohort, RSMR for hospitals

with the Centers for Medicare and Medicaid Services (CMS), the investigators are not able to make these data available to others; individual investigators interested in using these data should contact CMS directly. More information on data access can be found here: https://www.cms.gov/Research-Statistics-Data-and-Systems/CMS-Information-Technology/AccesstoDataApplication/index.html.

**Funding:** The analyses upon which this publication is based were performed under the Measure & Instrument Development and Support (MIDS) contract # HHSM-500-2013-13018I, Task Order HHSM-500-T0001, entitled Development, Reevaluation, and Implementation of Outcome/Efficiency Measures for Hospital and Eligible Clinicians, funded by the Centers for Medicare & Medicaid Services, an agency of the US Department of Health and Human Services. The content of this publication does not necessarily reflect the views or policies of the Department of Health and Human Services nor does the mention of trade names, commercial products, or organizations imply endorsement by the US government. The authors assume full responsibility for the accuracy and completeness of the ideas presented

**Competing interests:** EJB: None JG: None JR: In the past 36 months, Dr. Ross has received research support through Yale University from Johnson and Johnson to develop methods of clinical trial data sharing, from Medtronic, Inc. and the Food and Drug Administration (FDA) to develop methods for postmarket surveillance of medical devices (U01FD004585), from the Food and Drug Administration to establish Yale-Mayo Clinic Center for Excellence in Regulatory Science and Innovation (CERSI) program (U01FD005938), from the Blue Cross Blue Shield Association to better understand medical technology evaluation, from the Centers of Medicare and Medicaid Services (CMS) to develop and maintain performance measures that are used for public reporting (HHSM-500-2013-13018I), from the Agency for Healthcare Research and Quality (R01HS022882), from the National Heart, Lung and Blood Institute of the National Institutes of Health (NIH) (R01HS025164), and from the Laura and John Arnold Foundation to establish the Good Pharma Scorecard at Bioethics International and to establish the Collaboration for Research Integrity and Transparency (CRIT) at Yale. TA: None SP: None ND: None.

without versus with heart transplantation patients was 14.7% and 13.9% and RSRR was 17.8% and 17.7%, respectively; in the HF cohort, RSMR was 11.9% and 11.0%, and RSRR was 22.6% and 22.6%, respectively. Estimations changed ≤0.1% after excluding LVAD or heart transplantation patients.

## Conclusion

Hospitals caring for ≥6 patients with LVAD or ≥1 heart transplantation typically had a trend toward lower RSMRs but higher RSRRs. Rates were insignificantly changed when these patients were excluded. LVADs were primarily for acute-care in the AMI cohort and chronic support in the HF cohort. LVAD and heart transplantation patients are a distinct group with differential care requirements and outcomes, thus should be considered separately from the rest of the HF cohort.

## Introduction

LVADs and heart transplantations represent two advanced therapies for management of AMI or HF. Devices have different indications depending on many patient level factors, including the estimated length of time needed to support the failing heart and etiology of disease, among others.[1] Some devices are delivered percutaneously for temporary hemodynamic support (i.e. Impella® and TandemHeart® devices at the time of the study), while others are implanted for intermediate to longer term support as destination therapy or bridge to transplant (i.e. HeartMate® II, at the time of our study). Use of LVADs has been increasing, particularly for their use as destination therapy,[2] but also for temporary mechanical support.[3] Whereas, heart transplantation is reserved for patients refractory to any other therapies,[4] the frequency of which has recently remained stable over time due to lack of available donors.[5, 6]

Hospital 30-day readmission and mortality rates for acute myocardial infarction (AMI) and heart failure (HF) patients have been publicly reported as performance measures and incorporated into payment programs, such as the Hospital Readmission Reduction Program and the Hospital Value Based Purchasing program.[7, 8] Some have raised concerns about the appropriateness of including patients who receive LVAD devices in the measure cohorts due to the distinct clinical care and monitoring these patients require and their increased risk for some outcomes compared with other AMI and HF patients.[9] Differential consideration of this population has precedent; HF patients receiving LVAD or heart transplantation were found to have higher median payments and were excluded from 30-day episode of care heart failure payment measures for CMS.[10]

Prior to 2016, publicly reported 30-day all-cause mortality and readmission measures used by the Centers for Medicare and Medicaid Services (CMS) for patients admitted with AMI or HF included patients who received a LVAD or heart transplantation during index hospital admission.[11] We sought to quantify the available data on LVAD and heart transplant on AMI and HF mortality and readmission measures and to determine the impact of excluding them from the measures. Exclusion of LVAD and heart transplant patients will inform whether these patients significantly alter hospital level mortality and readmissions, thus informing stakeholder and policymakers on the implications of having them included or excluded from future measures. Reporting on these findings is relevant to current practice because these data were utilized by CMS in 2016 to determine inclusion and exclusion rules for LVAD and heart transplant patients from 30-day mortality and readmission measures.

## Materials and methods

### Data

We used data from Medicare Fee-For-Service beneficiaries that were discharged from July 2010 through June 2013. CMS uses 3 years of data for calculation of the HF and AMI 30-day mortality and readmission measures. Study cohorts were defined consistent with CMS methods for public reporting based on ICD-9CM codes for discharges from hospitals for Medicare beneficiaries aged 65 years or older with a principal diagnosis of AMI or HF.[12–15] Within each measure cohort, hospitalizations were assigned to one of four cohorts: LVAD or non-LVAD recipients and heart transplant recipients and non-recipients. Inclusion in the cohorts was defined by ICD-9CM procedure codes 37.60, 37.62, 37.65. 37.66. and 37.68 for the LVAD cohort and 33.6 and 37.51 in the heart transplant cohort. ICD-9CM codes for external temporary devices are designated by 37.60, 37.62, 37.65, and 37.68 and for implanted semi-permanent or permanent devices by 37.66. This population did not include patients with total artificial hearts.

### Outcome measures

The outcomes were 30-day, risk standardized mortality and readmission. Risk standardized measures seek to adjust for case mix differences between hospitals based on comorbidities and clinical status at the time of index admission. Additional analyses were performed to include cases up to one year prior to the index admission and after excluding the most common ICD-9CM code corresponding to external temporary LVADs.(37.68)

### Statistical analysis

First, we calculated the frequency of patients who received an LVAD or heart transplant in each of the four measure cohorts. We then separately calculated the frequencies of the use of external temporary and implanted semi-permanent or permanent devices according to their corresponding ICD-9CM codes. ICD-9CM code 37.66 corresponds to LVADs for semi-permanent or permanent use, whereas 37.60, 37.62, 37.65. and 37.68 corresponds to those for external/transient use. Unadjusted 30-day mortality and 30-day readmission rates at the patient level were then determined for patients who did and did not receive a LVAD or heart transplantation during the index admission.

We then estimated 30-day risk-standardized mortality rates (RSMR) and 30-day risk-standardized readmission rates (RSRR) for various groups at the hospital level. RSMR or RSRR take into account the hierarchical structure of the data to account for patient clustering within hospitals. Models include multiple covariates as well as hospital-specific random effects intercept.[14, 15] The RSMR or RSRR is calculated as a ratio of the number of "predicted" outcomes to the number of "expected" outcomes (death or readmission), multiplied by the national adjusted rate of the given outcome. For each hospital, the numerator of the ratio is the outcome within 30 days predicted on the bases of the hospital's performance within its observed case mix and the denominator is the expected number of outcomes on the basis of the performance of the nation's "average" hospital within this hospital's case mix.[16] Hospitals were stratified into groups based on the number of LVAD or heart transplant patients within their AMI or HF mortality and readmission cohorts over the time period. For LVAD, hospitals were stratified as none, at least one, and 6 or more LVAD patients. For heart transplantation, hospitals were stratified as having none or at least one heart transplantation patient.

For hospitals with at least one LVAD patient, 30-day RSMR and RSRR were re-estimated after excluding patients who received an LVAD during the index hospitalization to examine

the impact of this exclusion on measure cohorts. Due to the low frequency of heart transplantation patients, 30-day RSMR and RSRR were not recalculated after excluding heart transplantations at hospitals with at least one heart transplant patient or recalculated after combining LVAD and heart transplant patients. In a final analysis of the HF cohort, estimates were repeated for patients who received an LVAD or heart transplantation during the index admission with addition of those who also received one in the year prior to admission.

Observed readmission and mortality rates were reported as total number of patients and percent experiencing the clinical event. 30-day RSMR and RSRR are reported as percentiles (Median and interquartile range (IQR)) when comparing hospitals stratified by number of LVAD or heart transplantation patients. When analyses were repeated to exclude LVAD patients, 30-day median RSMR and RSRR are compared.

Due to the small numbers of patients with LVAD and heart transplantation in the cohorts, the study was not powered to detect differences and thus no statistical tests of significance were used. All analyses were done with SAS version 9.2 (SAS Institute Inc., Cary, NC). This work was exempted by the Yale University Human Investigation Committee. We obtained a waiver from institutional review board review by Yale University. Consent was not obtained for this study and was waived by the institutional review board since this study did not include the primary collection of data and utilized previously collected data from CMS.

## Results

### LVAD and transplant index admissions

LVAD and heart transplantation patients represented a small portion of the overall measure cohort. The AMI mortality cohort (n = 506,543 patients) had 1,166 patients (0.23%) who received an LVAD during index admission and 3 patients (<0.01%) who received a heart transplantation. For the AMI readmission cohort (n = 526,309 patients), there were 1,016 patients (0.19%) who received an LVAD during index hospital admission and 3 patients (<0.01%) who received a heart transplantation.

For the HF mortality cohort (n = 1,015,335 patients), there were 789 patients (0.08%) who received an LVAD during index admission and 212 patients (0.02%) who received a heart transplantation. For the HF readmission cohort (n = 1,254,124 patients) there were 931 patients (0.07%) who received an LVAD during index hospital admission and 220 patients (0.02%) who received a heart transplantation.

### Type of LVAD used

The AMI readmission and mortality cohorts most frequently used ICD-9CM codes that represented indications for external temporary placed LVADs at 97.2% and 98.2%, respectively. The HF cohorts differed in frequency of ICD-9CM codes used. The HF readmissions and mortality cohorts most frequently used ICD-9CM codes that correspond to placement of implanted semi-permanent or permanent LVADs at 78.0% and 73.8%, respectively. See Table 1 for frequency of ICD-9CM code utilization for AMI and HF mortality and readmission cohorts.

### Patient level observed mortality and readmission rates

Patients admitted for AMI who received an LVAD had higher unadjusted mortality and readmission rates than those who did not receive an LVAD (mortality: 42.5% (95% confidence interval (CI), 39.7%, 45.3%) vs. 14.8% (95% CI, 14.7%, 14.9%); readmission: 23.0% (95% CI, 20.4%, 25.6%) vs 17.8% (95% CI, 17.7%, 17.9%)). There were too few heart transplantation patients to make definitive comparisons on mortality and readmission in the AMI cohort.

**Table 1. Frequency of LVAD procedure codes in the AMI and HF readmission cohorts.**

**A. Frequency of LVAD procedure codes in the AMI readmission cohorts.**

| ICD-9CM Procedure Code | Frequency (n) | Rate (%) |
|---|---|---|
| External/temporary LVAD code(s) | 1186 | 97.2% |
| Implanted/Semi-permanent/ permanent LVAD code(s) | 28 | 2.8% |

**B. Frequency of LVAD procedure codes in the AMI mortality cohorts.**

| ICD-9CM Procedure Code | Frequency (n) | Rate (%) |
|---|---|---|
| External/temporary LVAD code(s) | 1145 | 98.2% |
| Implanted/Semi-permanent/ permanent LVAD code(s) | 21 | 1.8% |

**C. Frequency of LVAD procedure codes in the HF readmission cohorts.**

| ICD-9CM Procedure Code | Frequency (n) | Rate (%) |
|---|---|---|
| External/temporary LVADs | 205 | 22.0% |
| Implanted/semi-permanent/ permanent LVAD code(s) | 726 | 78.0% |

**D. Frequency of LVAD procedure codes in the HF mortality cohorts.**

| ICD-9CM Procedure Code | Frequency (n) | Rate (%) |
|---|---|---|
| External/temporary LVAD code(s) | 207 | 26.2% |
| Implanted/Semi-permanent/ permanent LVAD code(s) | 582 | 73.8% |

Patients admitted for HF who received an LVAD had similar mortality, but higher readmission rates than those who did not receive an LVAD (mortality:10.8% (95% CI, 8.6%, 13.0%) vs. 11.9% (95% CI, 11.8%,12.0%); readmission 32.8% (95% CI, 29.8%, 35.8%) vs 22.7% (95% CI, 22.6%, 22.8%). Patients admitted for HF who received a heart transplant had lower mortality, but similar readmission rates than those who did not receive a heart transplant (mortality 5.7% (95% CI, 2.6%, 8.8%) vs 11.9% (95% CI, 11.8%, 12.0%); readmission 18.6% (95% CI, 13.5%, 23.7%) vs 22.7% (95% CI, 22.6%, 22.8%). See Table 2 for complete comparisons.

## Risk-standardized mortality and readmission rates

For both the AMI and HF mortality and readmission cohorts, the vast majority of hospitals did not have any patients who received an LVAD. Less than 2% of hospitals had six or more Medicare patients who received an LVAD. Similar findings were also observed for hospitals that did or did not have heart transplantation patients, with less than 2% of hospitals performing heart transplantation.

For the AMI cohort, the median RSMRs for hospitals with 0, 1+, and 6+ LVAD patients were similar at 14.8% (IQR, 13.9, 15.8%), 14.4% (IQR, 13.4%, 15.6%), and 14.3% (IQR, 13.4%, 15.4%), respectively, while the median RSRRs were also similar at 17.8% (IQR, 17.2%, 18.5%), 18.0% (IQR, 17.1%, 18.9%), 18.3% (IQR, 17.4%, 18.9%), respectively. The median RSMRs for hospitals with no heart transplantations compared to at least one heart transplantation were 14.7% (IQR, 13.8%, 15.8%) and 13.9% (IQR, 12.4%, 15.1%), respectively, while the RSRRs were 17.8% (IQR, 17.2%, 18.6%) and 17.7% (IQR, 16.3%, 18.8%), respectively. See Table 3.

**Table 2. Patient level observed readmission and mortality rates with and without LVAD/transplant during the index admission.** Rates are unadjusted.

| Summary Statistics | AMI Mortality | | AMI Readmission | | HF Mortality | | HF Readmission | |
|---|---|---|---|---|---|---|---|---|
| | N | Rate | N | Rate | N | Rate | N | Rate |
| Non-LVAD patients | 505,377 | 14.8% | 525,293 | 17.8% | 1,014,546 | 11.9% | 1,253,193 | 22.7% |
| LVAD patients | 1,166 | 42.5% | 1,016 | 23.0% | 789 | 10.8% | 931 | 32.8% |
| Non-transplant patients | 506,540 | 14.9% | 526,306 | 17.8% | 1,015,123 | 11.9% | 1,253,904 | 22.7% |
| Transplant patients | 3 | 0.0% | 2 | 66.7% | 212 | 5.7% | 220 | 18.6% |

**Table 3. Acute myocardial infarction cohort RSMRs and RSRRs.**

**A. RSMR and RSRR for hospitals with 0, 1+, and 6+ LVAD patients.**

| Summary Statistics | Mortality | | | Readmission | | |
|---|---|---|---|---|---|---|
| | 0 LVADs N = 2183 | 1+ LVADs N = 432 | 6+ LVADs N = 34 | 0 LVADs N = 1989 | 1+ LVAD N = 389 | 6+ LVADs N = 22 |
| 25th Percentile | 13.9% | 13.4% | 13.4% | 17.2% | 17.1% | 17.4% |
| Median | 14.8% | 14.4% | 14.3% | 17.8% | 18.0% | 18.3% |
| 75th Percentile | 15.8% | 15.6% | 15.4% | 18.5% | 18.9% | 18.9% |

**B. RSMR and RSRR for hospitals without and with heart transplant patients.**

| Summary Statistics | Mortality | | Readmission | |
|---|---|---|---|---|
| | 0 transplant patients N = 2612 | 1+ transplant patients N = 3 | 0 transplant patients N = 2375 | 1+ transplant patients N = 3 |
| 25th Percentile | 13.8% | 12.4% | 17.2% | 16.3% |
| Median | 14.7% | 13.9% | 17.8% | 17.7% |
| 75th Percentile | 15.8% | 15.1% | 18.6% | 18.8% |

For the HF cohort, the median RSMRs for hospitals with 0, 1+, and 6+ LVAD patients were 11.9% (IQR, 11.0%, 13.0%), 11.2% (IQR, 10.0%, 12.4%), and 9.7% (IQR, 9.0%, 10.3%), respectively, while the median RSRRs were 22.6% (IQR, 21.7%, 23.7%), 22.6% (IQR, 21.3%, 24.2%), and 23.4% (IQR, 21.1%, 25.2%), respectively. The median RSMRs for hospitals with no heart transplantations compared to at least one heart transplantation were 11.9% (IQR, 11.0%, 13.0%) and 11.0% (IQR, 10.1%, 11.8%), respectively, while the median RSRRs were 22.6% (IQR, 21.7%, 23.7%), and 22.6% (IQR, 21.4%, 23.9%), respectively. See Table 4.

## Risk-standardized mortality and readmission rates after exclusion of LVAD patients

After exclusion of LVAD patients from the AMI cohort, the median RSMR and RSRR for hospitals with 1+ LVAD were not significantly different, with differences between medians of ≤0.1%. The results were similar after exclusion of LVAD patients from the HF cohort, with ≤0.1% difference from the prior median RSMR and RSRR for hospitals with 1+ LVAD.

**Table 4. Heart Failure cohort RSMRs and RSRRs.** Abbreviations: RSMR, risk-standardized mortality rate; RSRR, risk-standardized readmission rate.

**A. RSMR and RSRR for hospitals with 0, 1+, and 6+ LVAD patients.**

| Summary Statistics | Mortality | | | Readmission | | |
|---|---|---|---|---|---|---|
| | 0 LVADs N = 3829 | 1+ LVADs N = 146 | 6+ LVADs N = 11 | 0 LVADs N = 3950 | 1+ LVAD N = 143 | 6+ LVADs N = 15 |
| 25th Percentile | 11.0% | 10.0% | 9.0% | 21.7% | 21.3% | 21.1% |
| Median | 11.9% | 11.2% | 9.7% | 22.6% | 22.6% | 23.4% |
| 75th Percentile | 13.0% | 12.4% | 10.3% | 23.7% | 24.2% | 25.2% |

**B. RSMR and RSRR for hospitals with and without heart transplant patients.**

| Summary Statistics | Mortality | | Readmission | |
|---|---|---|---|---|
| | 0 transplant patients N = 3911 | 1+ transplant patients N = 64 | 0 transplant patients N = 4027 | 1+ transplant patients N = 66 |
| 25th Percentile | 11.0% | 10.1% | 21.7% | 21.4% |
| Median | 11.9% | 11.0% | 22.6% | 22.6% |
| 75th Percentile | 13.0% | 11.8% | 23.7% | 23.9% |

**Table 5. RSMRs and RSRRs for heart failure in the original cohort and after excluding patients receiving an LVAD or transplant during index admission or in the year prior (among all HF mortality and readmission hospitals).**

| Summary Statistics | RSMR | | RSRR | |
|---|---|---|---|---|
| | Original Cohort N = 3975 | Cohort with LVAD/Transplant patients excluded N = 3975 | Original Cohort N = 4093 | Cohort with LVAD/Transplant patients excluded N = 4093 |
| 25th Percentile | 11.0% | 11.0% | 21.7% | 21.7% |
| Median | 11.9% | 11.9% | 22.6% | 22.6% |
| 75th Percentile | 13.0% | 13.0% | 23.7% | 23.7% |

## Evaluation of the heart failure cohort with LVAD and heart transplant patients combined

After combining patients who received a LVAD or heart transplantation during the index admission or at any time in the year prior to admission and excluding ICD-9CM code 37.68 in the HF cohort, these patients again account for a minority of the patient populations (0.10% in the mortality cohort; 0.10% in the readmission cohort). The mortality rate of these combined patients in the HF cohort was 6.9% (95% CI, 5.3%, 8.5%) compared to the non-LVAD and non-heart transplantation patient rate of 12.0% (95% CI, 11.9%, 12.1%). The median hospital-level RSMR did not differ when including or excluding patients with LVAD or heart transplantation at 11.9% (IQR,11.0%, 13.0%). The readmission rate for this combined cohort was 30.6% (95% CI, 28.0%, 33.2%) compared to the non-LVAD and non-heart transplantation patient rate of 22.7% (95% CI, 22.6%, 22.8%). The median hospital-level RSRR was also the same when including or excluding patients with LVAD or heart transplant at 22.6% (IQR, 21.7%, 23.7). See Table 5.

When narrowed to hospitals with at least one LVAD or one transplant patient, RSMR for hospitals were unchanged with inclusion or exclusion of LVAD and heart transplant patients at 11.0% (IQR, 10.2%, 11.9%). The RSRR for this comparison was also similar at 22.6% (IQR, 21.4%, 24.0%) compared to 22.7% (IQR, 21.3%, 24.0%) when excluding such patients. See Table 6.

## Discussion

In this study, risk standardized outcome rates for hospitals that cared for LVAD or heart transplant patients tended to have slightly lower mortality and slightly higher readmission rates compared to those that did not care for such patients; differences were mostly less than 1% and in many cases less than 0.5%. Hospitals caring for LVAD and transplant patients do not have substantially different risk-standardized outcome rates than other hospitals likely because LVAD and transplant patients represent a small proportion of hospitals' overall patients and because much of the difference in observed outcome rates is explained by comorbidities and

**Table 6. RSMRs and RSRRs for heart failure in the original cohort and after excluding patients receiving an LVAD or transplant during index admission or in the year prior (among hospitals with at least one LVAD/transplant patients).** Abbreviations: RSMR, risk-standardized mortality rate; RSRR, risk-standardized readmission rate.

| Summary Statistics | Mortality | | Readmission | |
|---|---|---|---|---|
| | Original Cohort N = 148 | Cohort with LVAD/Transplant patients excluded N = 148 | Original Cohort N = 176 | Cohort with LVAD/Transplant patients excluded N = 176 |
| 25th Percentile | 10.2% | 10.2% | 21.4% | 21.3% |
| Median | 11.0% | 11.0% | 22.6% | 22.7% |
| 75th Percentile | 11.9% | 11.9% | 24.0% | 24.0% |

severity of illness in these patients that are captured in the risk-standardized models. Thus, exclusion of LVAD and heart transplant patients resulted in ≤0.1% change in results. Notably, indication for LVAD use in the AMI and HF populations greatly differed between these two populations. This suggests that LVAD and heart transplantation are employed in disparate clinical scenarios and should be considered separately. The data we present here were the primary considerations in the CMS decision to exclude LVAD and heart transplant patients from HF, but not AMI, mortality and readmission measures.

In all of the cohorts evaluated, RSMR was always lower among hospitals that cared for LVAD or heart transplant patients. The fact that exclusion of such patients from the measure did not result in differences more than 0.1%, suggests that the observed differences are either a result of differential case mix or hospital care. The utilized model adjusts for case mix, thus making this less likely the cause for the differences and suggests centers capable of LVAD may have other factors that contribute to lower mortality aside from LVAD use.

The results of the RSRR can help us better understand these differences. Across all cohorts, RSRR were slightly higher in the LVAD cohorts and ≤0.1% different in the heart transplant cohorts. Again, exclusion of LVAD and heart transplant patients changed these values by ≤0.1%. Thus, hospitals that cared for LVAD and heart transplant patients achieved lower RSMR despite slightly higher RSRR. Again, since our model corrects for case mix, this again supports there being differences in hospital level care required to achieve a lower mortality rate despite a population with a higher readmission rate. This differential overall is interpreted as better outcomes from hospitals that care for LVAD and heart transplant patients that are independent on the utilization of LVAD or heart transplantation. Part of these difference is likely due to these hospitals having other unmeasured factors that allow them to handle the complexity of care required for LVAD and heart transplant patients.

These data were utilized by CMS to update mortality and readmission measures. Although the proportion of patients that received advanced therapies remains a small proportion and a minority of hospitals use these therapies, key considerations at the patient and hospital level were important for this decision. The clinically different scenarios and treatment goals that elicit use of LVAD and heart transplant in AMI as compared with HF plus their differential use in these populations and differences in unadjusted and risk adjusted mortality and readmission rates were all used by CMS for whether to continue including LVAD and heart transplant patients in reported measures. In 2016, CMS updated mortality and readmission measures to continue including LVAD and heart transplantation patients in AMI measures, but exclude LVAD and heart transplantation patients from HF measures.[17, 18]

Measuring and publicly reporting hospital level performance measures requires diligence to ensure that these measures accurately reflect patient care at the hospital level. This includes carefully adjusting for disease severities and case heterogeneity between hospitals. As we continue to understand these contexts, measures should continue to be revised in response.

Our study also has limitations. This study utilized a non-matched cohort and this limits interpretation of mortality and readmission rates between groups obtaining LVAD or heart transplantation. Also, our study was limited to Medicare patients only and did not include patients that have commercial insurance only. Results should be interpreted in this context. Additionally, since data were not available as to type of LVAD utilized, comparisons could not be made between types of LVAD (i.e. centrifugal vs. axial vs. percutaneously inserted devices). Furthermore, ICD-9CM code 37.66 also codes for right ventricular assist devices, thus there were likely some individuals with these devices included in the cohort. Although, RVADs are clinically less common and were likely rare events. This study also did not include the ICD-9CM code for total artificial hearts.(37.52) Notably, the data were from 2010–2013, which reflects outcomes from an older generation of LVAD devices. More recent data were not

available to the authors for the purposes of this study. Furthermore, these data did not allow for a determination of whether the indication for use was semi-permanent or permanent, as in, whether devices were intended as bridge-to-transplant or destination therapy. Thus, evaluating our findings in the context of these clinical goals could not be discussed. In addition, more robust hospital level data were not available to compare hospital level factors that may have contributed to mortality or readmission between hospitals that were and were not utilizers of heart transplantation or LVAD.

## Conclusion

In conclusion we report that hospitals caring for patients with LVAD or performed heart transplants typically had a trend toward lower RSMR but higher RSRR, which were not significantly changed when these patients were excluded. These differences are likely a result of hospital level differences in care that amount to better mortality result in hospitals capable of caring for LVAD and transplant patients. In 2016, CMS decided to exclude LVAD and transplant patients from HF mortality and readmission measures, due in part to these findings. Continued assessment of select patient populations can inform on the optimal populations to track for AMI and HF mortality and readmission measures.

## Author Contributions

**Conceptualization:** Susannah M. Bernheim, Nihar R. Desai.

**Formal analysis:** Joseph S. Ross, Jacqueline N. Grady, Nihar R. Desai.

**Investigation:** Joseph S. Ross, Jacqueline N. Grady, Tariq Ahmad, Susannah M. Bernheim, Nihar R. Desai.

**Methodology:** Joseph S. Ross, Nihar R. Desai.

**Project administration:** Joseph S. Ross, Jacqueline N. Grady, Tariq Ahmad, Susannah M. Bernheim, Nihar R. Desai.

**Resources:** Susannah M. Bernheim, Nihar R. Desai.

**Supervision:** Tariq Ahmad, Susannah M. Bernheim, Nihar R. Desai.

**Validation:** Nihar R. Desai.

**Visualization:** Susannah M. Bernheim, Nihar R. Desai.

**Writing – original draft:** Eric J. Brandt, Joseph S. Ross, Jacqueline N. Grady, Tariq Ahmad, Sumeet Pawar, Susannah M. Bernheim, Nihar R. Desai.

**Writing – review & editing:** Eric J. Brandt, Joseph S. Ross, Jacqueline N. Grady, Tariq Ahmad, Sumeet Pawar, Susannah M. Bernheim, Nihar R. Desai.

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
