## [Decision Letter · Decision Letter 0]

28 Nov 2019

PONE-D-19-30581

Impact of left ventricular assist devices and heart transplant on acute myocardial infarction and heart failure mortality and readmission measures

PLOS ONE

Dear Dr. Desai,

Thank you for submitting your manuscript to PLOS ONE. After careful consideration, we have decided that your manuscript does not meet our criteria for publication and must therefore be rejected.

Specifically, there were concerns whether this study is novel or much additive to the knowledge in the field, while the conclusions were deemed vague and the clinical implication of these results questionable.

I am sorry that we cannot be more positive on this occasion, but hope that you appreciate the reasons for this decision.

Yours sincerely,

Vakhtang Tchantchaleishvili

Academic Editor

PLOS ONE

Reviewers' comments:

Reviewer #1: The authors conduct an interesting study examining the rate of risk standardized mortality rates and readmission rates in an acute myocardial infarction and heart failure population with respect to heart failure and LVAD patients.

Abstract comments: In results section would specify exact number, less than or equal to 0.2% does not make sense in this context. Same with less than or equal to 0.1%. How could the number of patients be less than or equal to? Shouldn’t this be a whole integer? Would refer to semi-permanent/permanent as the indication for the LVAD, either bridge-to-transplant (semi-permanent) or destination therapy (permanent)

Introduction: Drawing similarities between devices like CF-LVADs and percutaneous support devices is rather heterogeneous. The indications of these devices and devices are far too different to conflate them. May be better to state actual devices (Impella for percutaneous support, Heartmate II and III for durable devices). Later in the intro paragraph, these are again conflated. When CF-LVAD or LVAD is written, most people would default to an assumption of a durable CF-VAD. The authors should clarify the distinction between percutaneous and durable devices.

Materials and methods: The inclusion of CF-LVAD and percutaneous support devices needs to be acknowledged in the limitations since they have different indications. This is partially acknowledged in the limitations section. Durable CF-LVAD patients and percutaneous support patients are not exactly similar. For the heart transplant patients, these are patients who were transplanted at this index hospitalization for heart failure?

Discussion:

Page 17, “supports their being difference” should be corrected to there

General comments: Can the authors clarify if these were pulsatile flow LVADs or continuous-flow? Given the contemporary cohort, it would be assumed that the LVAD population is continuous flow. Given that, the authors should replace LVAD at each instance with continuous-flow LVAD or CF-LVAD. Did this population include patients who had RVAD devices placed? Were any total artificial hearts included in the cohort?

Limitation: by including only medicare patients a significant number of patients are not included in the cohort some recent studies of 200 patients cite a mean age of 59. CF-LVAD patients have both a destination therapy and bridge-to-transplant indication. Did this study include both patient populations?

Reviewer #3: In this retrospective analysis of Medicare free-for-service beneficiaries with primary diagnosis of acute myocardial infarction (AMI) or heart failure (HF), Brandt et al presented the effects of including LVAD and heart transplant (HT) patients on 30-day readmission and mortality measures. The authors found the including or excluding these patients of patients does not significantly impact these measures though there were more readmission and less mortality among patients implanted an LVAD or underwent HT. The authors then concluded that LVAD and HT patients are still a distinct group with different care requirements and outcomes and should be analyzed separately from other groups of patients with HF/AMI. The manuscript is well written and methods are appropriate. However, these findings are largely expected and not novel. The clinical implication is questionable as the number of patients with HT or LVAD is very small relative to the large group of patients with chronic HF or AMI.

Reviewer #4: The authors of “Impact of left ventricular assist devices and heart transplant on acute myocardial infarction and heart failure mortality and readmission measures” speak to whether or not exclusion of LVAD and heart transplant patients significantly change the outcomes of CMS public reporting readmission and mortality for acute MI and heart failure patients. The statistics are done rigorously, and the manuscript is well-written in easy to understand English.

The authors utilized modeling similar to past publications to calculate the risk standardized readmission rate and mortality rate for these hospitals, and ultimately compared whether or not these values changed when LVAD and HT patients were excluded from the cohort. In the abstract, the authors state that their data supports that HT and LVAD patients should be considered separately from the all-comer HF patient population, although the fact that the exclusions do not change the RSMR and RSRR does not support that conclusion. Furthermore, the data show a similar lack of change in RSMR and RSRR in the AMI, which the authors do not discuss. In the discussion of the paper, the authors state that there is a dichotomy in AMI patients vs HF patients, in that the indication for LVAD shows that the AMI patients had external LVADs placed, vs HF patients receiving durable LVAD options, and that this likely drove the decision for the CMS to exclude HT and LVAD patients from their HF data, but not AMI data. In the discussion the authors do not really state whether or not they believe that HT and LVAD patients should be excluded from the HF and AMI groups, they simply discuss what the CMS policy reflects. Overall this discussion is confusing because the abstract and discussion of the paper do not deliver a consistent message, it would be helpful for the reader if the authors discussed why they believe their findings do/do not support the CMS policy decisions.

Furthermore, it is hard to believe that the exclusion of the LVAD patients and HT patients would affect the RSMR and RSRR values for the entire cohort, these exclusions make up ~1,000/1,000,000 of the HF patients (1/1000th of the cohort) and 1/500th of the AMI group. This is not discussed by the authors, it seems as though it is statistically impossible for the exclusion of such a small group to meaningfully shift the entire cohort’s mortality/morbidity.

Some points for consideration

-What was the purpose of stratifying LVADs into three groups and HT in to two groups? A discussion of this would be helpful to the reader

- Table 3B has incredibly small sample size for hospitals with more than 1 transplant. All data regarding heart transplant on AMI patients is based on an incredibly small sample size. It is unlikely that these data hold up across all hospitals in the united states or abroad considering that there are only three hospitals that are doing transplant in this cohort. Also, the clinical track to receiving a transplant in the setting of an acute MI is not entirely clear.

- In table 3a, 3b, 4a and 4b Why are the N values for hospitals with hospitals with and without HT/LVAD patients different in terms of mortality and readmissions? Shouldn’t the N value for these hospitals be equal in terms of hospitals that had heart failure patients, that were considered for readmission/mortality data? Were hospitals that have no deaths and no readmissions excluded from the reported N? For example, there is a discrepancy of over 100 hospitals between mortality and readmission hospitals with no transplants.

-The authors state that “Hospitals caring for >6 patients with LVAD or >1 HT had lower RSMRs but higher RSRRs on average.” It would be helpful to know what confidence intervals there are for this methodology. Most of the differences are less than 1% (one is equal, in fact), are we confident the modeling can say that these are truly different?

-It is worth adding a discussion regarding the shortcomings of using the database from 2010 to 2013. This data is being derived from the previous generation of LVADs (at least from a durable LVAD perspective). The authors should acknowledge that all of the data presented in this paper regarding devices are not commonly being used at this time (Impellas have been redesigned, Heartmate 3, and HVAD have been approved since the study period). Overall, it is not clear why the authors chose to use the CMS data from 2010-2013, when they feasibly could have used 2013-2016 (up until the date that the CMS changed their reporting guidelines). It would be helpful to know why the authors chose these dates; at face value it does not seem like these are the most recent data available to the authors. The fact that all most LVADs that are used today are a newer generation from what is being studied in this paper should be discussed by the authors.

6. PLOS authors have the option to publish the peer review history of their article (what does this mean?). If published, this will include your full peer review and any attached files.

- - - - -

---

## [Author Response · Author response to Decision Letter 0]

27 Dec 2019

Please note that this information can also be found in the attached file uploaded for the submission.

ONE-D-19-30581: Impact of left ventricular assist devices and heart transplant on acute myocardial infarction and heart failure mortality and readmission measures

Editor’s comments:

Thank you for submitting your manuscript to PLOS ONE. After careful consideration, we have decided that your manuscript does not meet our criteria for publication and must therefore be rejected. Specifically, there were concerns whether this study is novel or much additive to the knowledge in the field, while the conclusions were deemed vague and the clinical implication of these results questionable. I am sorry that we cannot be more positive on this occasion, but hope that you appreciate the reasons for this decision.

Response: We thank the editor for his consideration of our paper and that the reviewers were provided a timely response. However, we were confused as to why the decision was to reject. Our manuscript clearly meets the criteria for publication in PLOS ONE as outlined on the journal website. Most importantly we disagree with the assertion that the study is not novel or does not add to the field. This study has not been previously published nor the hypothesis tested previously. While the findings may be “expected,” our understanding was that part of the purpose of PLOS ONE was to publish research that is scientifically valid and meets methodological and ethical standards – not perceived significance. That said, the subject of our study, the impact of inclusion of LVAD and heart transplant patients on hospital measures of healthcare quality, has been a concern for key stakeholders and therefore represents an important and significant topic. We feel our study is both important and significant since they were the primary data that CMS used to make the decision of including or excluding these individuals from mortality and readmission measures. These measures remain the focus of national debate and continue to be key measures for hospital quality. Lastly, as had been pointed out by reviewers, our study was performed with high technical standards and are presented clearly. Therefore, since we feel our article meet all criteria and represent important findings we request an appeal for the decision to reject this manuscript.

Thank you for your reconsideration.

Reviewers' comments:

Reviewer #1: The authors conduct an interesting study examining the rate of risk standardized mortality rates and readmission rates in an acute myocardial infarction and heart failure population with respect to heart failure and LVAD patients.

1. Abstract comments: In results section would specify exact number, less than or equal to 0.2% does not make sense in this context. Same with less than or equal to 0.1%. How could the number of patients be less than or equal to? Shouldn’t this be a whole integer? Would refer to semi-permanent/permanent as the indication for the LVAD, either bridge-to-transplant (semi-permanent) or destination therapy (permanent)

Response: Thank you for these suggestions. We agree that it will help readers have a better understanding of the size of the cohort, as well as the number with devices, by providing this information in the abstract. In addition, we have considered the change in phrasing from semi-permanent/permanent to bridge-to-transplant or destination therapy. However, after discussion among the team, we respectively feel it is best to utilize the original terms. This will help to prevent misunderstanding regarding the intentions of the LVAD, since this cannot be necessarily construed from the indication. Nevertheless, in response to this comment, we have updated the abstract text to include the size of the cohort and number of devices in the abstract as follows (page 2): “In the AMI mortality (n=506,543) and readmission (n=526,309) cohorts, 1,166 and 1,016 patients received an LVAD while 3 and 2 received HT, respectively. In the HF mortality (n=1,015,335) and readmission (n=1,254,124) cohorts, 789 and 931 received and LVAD, while 212 and 202 received HT, respectively.”

2. Introduction: Drawing similarities between devices like CF-LVADs and percutaneous support devices is rather heterogeneous. The indications of these devices and devices are far too different to conflate them. May be better to state actual devices (Impella for percutaneous support, Heartmate II and III for durable devices). Later in the intro paragraph, these are again conflated. When CF-LVAD or LVAD is written, most people would default to an assumption of a durable CF-VAD. The authors should clarify the distinction between percutaneous and durable devices.

Response: We agree with the reviewer that CF-LVADs and percutaneous support devices often have different indications. Percutaneous devices are much more likely to be used in acute care scenarios such as AMI whereas CF-LVADs are better suited for chronic scenarios such as chronic HF. Ideally we would have known the exact devices used in each scenario. However, these data were not available within the dataset since the administrative codes were not granular enough to identify a specific device. However, this limitation does not detract from the primary purpose of this study, which was to understand the impact of including patient that receive heart transplant or any LVAD type devices on mortality and readmission measures. Furthermore, we did use available data on device indication to guide interpretation of our analyses. The fact that LVADs were primarily indicated for acute care in AMI but for chronic care in HF suggests that, as this reviewer hints, percutaneous devices are more often employed in the former verse the latter scenario. We include these points within the discussion, which also responds to concerns from Reviewer #4 (page 16): “Notably, indication for LVAD use in the AMI and HF populations greatly differed between these two populations. This suggests that LVAD and HT are employed in disparate clinical scenarios and should be considered separately;” We have also recognized the specific limitation this reviewer points out (page 18): “Additionally, since data were not available as to type of LVAD utilized, comparisons could not be made between types of LVAD (i.e. centrifugal vs. axial vs. percutaneously inserted devices).”

3. Materials and methods: The inclusion of CF-LVAD and percutaneous support devices needs to be acknowledged in the limitations since they have different indications. This is partially acknowledged in the limitations section. Durable CF-LVAD patients and percutaneous support patients are not exactly similar. 

Response: The reviewer is correct in pointing out that these devices have distinct indications. As mentioned in the above response, we would ideally be able to separate out the indication and determine which exact device was implanted and in which exact clinical scenario. These data do not allow for this type of evaluation. As such we have expanded our limitations to address this issue. Please note that at other points in the text we are sure to point out the importance in type of LVAD and how that was important for deciding which devices to include or exclude in measures. We have update the text as follow, which also helps to address concerns from response #1 (page 18): “Furthermore, these data did not allow for a determination of whether the indication for use was semi-permanent or permanent, as in, whether devices were intended as bridge-to-transplant or destination therapy. Thus, evaluating our findings in the context of these clinical goals could not be discussed.”

4. For the heart transplant patients, these are patients who were transplanted at this index hospitalization for heart failure?

Response: Yes, that is correct, patients in these respective cohorts received devices or transplant during the index hospitalization.

5. Discussion: Page 17, “supports their being difference” should be corrected to there

Response: This has been correct in the text to read as follows (page 17): “…supports there being differences…”.

6. General comments: Can the authors clarify if these were pulsatile flow LVADs or continuous-flow? Given the contemporary cohort, it would be assumed that the LVAD population is continuous flow. Given that, the authors should replace LVAD at each instance with continuous-flow LVAD or CF-LVAD.

Response: These data do not provide data on the level of understanding types of LVADs that were implanted beyond the indication. Given that the type of LVAD implanted is not adjudicated from these data we feel it would be inaccurate to make this assumption that they were all continuous flow LVADs. We have pointed this out in the manuscript with the following sentence of the limitations (page 18): “Additionally, since data were not available as to type of LVAD utilized, comparisons could not be made between types of LVAD (i.e. centrifugal vs. axial vs. percutaneously inserted devices).”

7. Did this population include patients who had RVAD devices placed? Were any total artificial hearts included in the cohort?

Response part 1: The ICD-9CM code specifying RVAD fall under 37.66, which were included in our cohort definition. We expect based on clinical practice that these were a minority of the population sampled. To reflect this limitation, we have included a statement in the limitations (page 18): “Furthermore, ICD-9CM code 37.66 also includes right ventricular assist devices, thus there were likely some individuals with these devices included in the cohort. Although, these are clinically less common and were likely rare events.”

Response part 2: This population did not include patients with total artificial hearts placed. Text was added to the methods and limitations to clarify this point (page 5): “This population did not include patients with total artificial hearts.” See also (page 18): “This study also did not include the ICD-9CM code for total artificial hearts (37.52).”

8. Limitation: by including only Medicare patients a significant number of patients are not included in the cohort some recent studies of 200 patients cite a mean age of 59. CF-LVAD patients have both a destination therapy and bridge-to-transplant indication. Did this study include both patient populations?

Response: The reviewer is correct that our study was limited to Medicare patients, but not individuals with commercial or other types of insurance. As such, the reviewer is correct that if the mean age of LVAD implantation is below the typical Medicare age then majority of the patients for which this applies were not included in the study. However, because the purpose of our study was to understand the impact of including these patients on the measures used by Medicare for purposes of characterizing hospital quality of care (which are limited to patients enrolled in fee-for-service Medicare), there would be no reason to include patients who did not have Medicare insurance. Nevertheless, to be responsive to the Reviewer’s concern, we have updated our manuscript to reflect that this study was limited to only include Medicare patients and therefore was not inclusive of the total population (page 18): “Also, our study was limited to Medicare patients only and did not include patients that have commercial insurance only. Results should be interpreted in this context.”

Reviewer #3:

1. In this retrospective analysis of Medicare free-for-service beneficiaries with primary diagnosis of acute myocardial infarction (AMI) or heart failure (HF), Brandt et al presented the effects of including LVAD and heart transplant (HT) patients on 30-day readmission and mortality measures. The authors found the including or excluding these patients of patients does not significantly impact these measures though there were more readmission and less mortality among patients implanted an LVAD or underwent HT. The authors then concluded that LVAD and HT patients are still a distinct group with different care requirements and outcomes and should be analyzed separately from other groups of patients with HF/AMI. The manuscript is well written and methods are appropriate. However, these findings are largely expected and not novel. The clinical implication is questionable as the number of patients with HT or LVAD is very small relative to the large group of patients with chronic HF or AMI.

Response: We thank the reviewer for his complement that our manuscript is well written with appropriate methods. However, we disagree with the assertion that the findings are not novel and lack implications. First, these findings have not been previously published in the literature and, as such, are novel. Second, this study has direct policy relevance, which in turn gives it great clinical relevance, since it was used by CMS to decide on inclusion and exclusion criteria for the mortality and readmission measures used by the agency to characterize hospital quality of care. Adding these findings to the literature provides scientific merit to the decisions as well as adds the scientific discourse on how these measures are determined. 

Reviewer #4:

1. The authors of “Impact of left ventricular assist devices and heart transplant on acute myocardial infarction and heart failure mortality and readmission measures” speak to whether or not exclusion of LVAD and heart transplant patients significantly change the outcomes of CMS public reporting readmission and mortality for acute MI and heart failure patients. The statistics are done rigorously, and the manuscript is well-written in easy to understand English.

Response: We thanks this reviewer for the complement

2. The authors utilized modeling similar to past publications to calculate the risk standardized readmission rate and mortality rate for these hospitals, and ultimately compared whether or not these values changed when LVAD and HT patients were excluded from the cohort. In the abstract, the authors state that their data supports that HT and LVAD patients should be considered separately from the all-comer HF patient population, although the fact that the exclusions do not change the RSMR and RSRR does not support that conclusion. Furthermore, the data show a similar lack of change in RSMR and RSRR in the AMI, which the authors do not discuss. In the discussion of the paper, the authors state that there is a dichotomy in AMI patients vs HF patients, in that the indication for LVAD shows that the AMI patients had external LVADs placed, vs HF patients receiving durable LVAD options, and that this likely drove the decision for the CMS to exclude HT and LVAD patients from their HF data, but not AMI data. In the discussion the authors do not really state whether or not they believe that HT and LVAD patients should be excluded from the HF and AMI groups, they simply discuss what the CMS policy reflects. Overall this discussion is confusing because the abstract and discussion of the paper do not deliver a consistent message, it would be helpful for the reader if the authors discussed why they believe their findings do/do not support the CMS policy decisions.

Response: We appreciate the Reviewer’s comment that our abstract and discussion text do not align in tone. We have looked closely at both sections of our manuscript and revised our language to ensure that it is clear that CMS is the ultimate decision maker of how to utilize the data for policy, as well as our own interpretations of the research, independent of CMS. As such, we have updated the discussion to parallel the tone of the abstract while still emphasizing CMS as the primary decision maker for defining inclusion criteria for mortality and readmission measures. We have updated the text as follows, which also touches on concerns from Reviewer #1 (page 16): “Notably, indication for LVAD use in the AMI and HF populations greatly differed between these two populations. This suggests that LVAD and HT are employed in disparate clinical scenarios and should be considered separately. The data we present here were the primary considerations in the CMS decision to exclude LVAD and heart transplant patients from HF, but not AMI, mortality and readmission measures.”

3. Furthermore, it is hard to believe that the exclusion of the LVAD patients and HT patients would affect the RSMR and RSRR values for the entire cohort, these exclusions make up ~1,000/1,000,000 of the HF patients (1/1000th of the cohort) and 1/500th of the AMI group. This is not discussed by the authors, it seems as though it is statistically impossible for the exclusion of such a small group to meaningfully shift the entire cohort’s mortality/morbidity.

Response: In regards to this observation, many stakeholders had noted concerns that despite making up a small portion of the population that they could have significant implications for individual hospitals. We did confirm that they are a small portion of the population but did not confirm concerns that they significantly alter performance. Notably we did bring attention to the small proportion of patients with LVAD and HT that make up the cohort. We were sure to point out that these patients make up a small portion of the cohort throughout the manuscript.

Some points for consideration:

4. What was the purpose of stratifying LVADs into three groups and HT in to two groups? A discussion of this would be helpful to the reader

Response: The purpose of the stratification scheme for LVADs was to see whether higher volume centers would be more affected. Due to the number of transplant being low we could only do two groups and not three.

5. Table 3B has incredibly small sample size for hospitals with more than 1 transplant. All data regarding heart transplant on AMI patients is based on an incredibly small sample size. It is unlikely that these data hold up across all hospitals in the united states or abroad considering that there are only three hospitals that are doing transplant in this cohort. Also, the clinical track to receiving a transplant in the setting of an acute MI is not entirely clear.

Response: The reviewer is correct in pointing out that heart transplant as a result from AMI hospitalization is a rare event. A priori this could have been predicted. However, these data are important, despite the small counts, for considering populations to be included or excluded from admission and readmission measures. As mentioned above in response #3, this was important to for stakeholders to understand whether these patients alter performance measures.

6. In table 3a, 3b, 4a and 4b Why are the N values for hospitals with hospitals with and without HT/LVAD patients different in terms of mortality and readmissions? Shouldn’t the N value for these hospitals be equal in terms of hospitals that had heart failure patients, that were considered for readmission/mortality data? Were hospitals that have no deaths and no readmissions excluded from the reported N? For example, there is a discrepancy of over 100 hospitals between mortality and readmission hospitals with no transplants.

Response: The CMS inclusion and exclusion criteria used to define the mortality and readmission measures for the cohorts have differences. This results in slightly different values for hospitals that are included in the individual cohorts. Hospitals that had no patients meeting criteria were not included in the study cohorts. However, if a hospital had patients meeting criteria but these patients had no readmission or deaths then these were included.

7. The authors state that “Hospitals caring for >6 patients with LVAD or >1 HT had lower RSMRs but higher RSRRs on average.” It would be helpful to know what confidence intervals there are for this methodology. Most of the differences are less than 1% (one is equal, in fact), are we confident the modeling can say that these are truly different?

Response: The term on average could be construed as a statistical term. However, in this context we are referring to non-statistical comparisons of medians. As such we have corrected the text for the readers to understand that in these conclusions. We have updated the text remove the term “on average” and it now reads as follows in 2 locations (page 3): “Hospitals caring for >6 patients with LVAD or >1 HT typically had lower RSMRs but higher RSRRs.” (page 19): “In conclusion we report that hospitals caring for patients with LVAD or performed heart transplants typically had lower RSMR but higher RSRR, which were not significantly changed when these patients were excluded.”

8. It is worth adding a discussion regarding the shortcomings of using the database from 2010 to 2013. This data is being derived from the previous generation of LVADs (at least from a durable LVAD perspective). The authors should acknowledge that all of the data presented in this paper regarding devices are not commonly being used at this time (Impellas have been redesigned, Heartmate 3, and HVAD have been approved since the study period). Overall, it is not clear why the authors chose to use the CMS data from 2010-2013, when they feasibly could have used 2013-2016 (up until the date that the CMS changed their reporting guidelines). It would be helpful to know why the authors chose these dates; at face value it does not seem like these are the most recent data available to the authors. The fact that all most LVADs that are used today are a newer generation from what is being studied in this paper should be discussed by the authors.

Response part 2: It is noted that pointing out that newer devices may have differing outcomes. It is also worth noting that this is an important piece to continued efforts to study mortality and readmissions in relation to transplant and LVADs. The text has been updated to include the following text (page 18): “Notably, the data were from 2010-2013, which reflects outcomes from an older generation of LVAD devices. More recent data were not available to the authors for the purposes of this study.”

---

## [Decision Letter · Decision Letter 1]

5 Feb 2020

PONE-D-19-30581R1

Impact of left ventricular assist devices and heart transplant on acute myocardial infarction and heart failure mortality and readmission measures

PLOS ONE

Dear Dr. Desai,

Thank you for submitting your manuscript to PLOS ONE. After careful consideration, we feel that it has merit but does not fully meet PLOS ONE’s publication criteria as it currently stands. Therefore, we invite you to submit a revised version of the manuscript that addresses the points raised during the review process.

This manuscript has been evaluated by two academic editors and two external reviewers (different from the initial reviewers). We are in agreement with the Reviewer's request for major revisions based on the feedback received.

We would appreciate receiving your revised manuscript by Mar 21 2020 11:59PM. To enhance the reproducibility of your results, we recommend that if applicable you deposit your laboratory protocols in protocols.io, where a protocol can be assigned its own identifier (DOI) such that it can be cited independently in the future. For instructions see: http://journals.plos.org/plosone/s/submission-guidelines#loc-laboratory-protocols

We look forward to receiving your revised manuscript.

Kind regards,

Saraschandra Vallabhajosyula, MD FACP

Academic Editors

PLOS ONE

**Journal Requirements:**

2) We note that you have indicated that data from this study are available upon request. PLOS only allows data to be available upon request if there are legal or ethical restrictions on sharing data publicly. For information on unacceptable data access restrictions, please see http://journals.plos.org/plosone/s/data-availability#loc-unacceptable-data-access-restrictions.

3) Thank you for stating the following in the Competing Interests section:

"EJB: None

JG: None

JR: In the past 36 months, Dr. Ross has received research support through Yale University from Johnson and Johnson to develop methods of clinical trial data sharing, from Medtronic, Inc. and the Food and Drug Administration (FDA) to develop methods for postmarket surveillance of medical devices (U01FD004585), from the Food and Drug Administration to establish Yale-Mayo Clinic Center for Excellence in Regulatory Science and Innovation (CERSI) program (U01FD005938), from the Blue Cross Blue Shield Association to better understand medical technology evaluation, from the Centers of Medicare and Medicaid Services (CMS) to develop and maintain performance measures that are used for public reporting (HHSM-500-2013-13018I), from the Agency for Healthcare Research and Quality (R01HS022882), from the National Heart, Lung and Blood Institute of the National Institutes of Health (NIH) (R01HS025164), and from the Laura and John Arnold Foundation to establish the Good Pharma Scorecard at Bioethics International and to establish the Collaboration for Research Integrity and Transparency (CRIT) at Yale.

TA: None

SP: None

ND: None".

We note that one or more of the authors have an affiliation to the commercial funders of this research study : [Johnson and Johnson and Medtronic, Inc ].

**Additional Editor Comments (if provided):**

The authors intended to assess the impact of LVAD and HTx, a subgroup, on some outcome measures. These surgical cases were a small minority and stat. insig changed those rates. In epidemiology, it is always better to enhance homogeneity to reduce selection bias. So it was not a bad idea to exclude those cases. But since their existence did not affect the rates significantly by statistical testing, why not just keep them to reflect the reality in clinical scenario? The authors needed to clarify the purpose of those outcome measures. Is there any benefit to exclude LVAD and HTx, such as for cost management, admin, insurance, etc? Please revise to reflect the motivation of the manuscript.

LVAD and HTx, as a small minority, did not affect the outcome measures. Their non-exclusion did not violate homogeneity but also reflect the diversity of clinical reality. The authors need to provide a stronger reason to exclude them

Reviewers' comments:

Reviewer's Responses to Questions

**Comments to the Author**

1. If the authors have adequately addressed your comments raised in a previous round of review and you feel that this manuscript is now acceptable for publication, you may indicate that here to bypass the “Comments to the Author” section, enter your conflict of interest statement in the “Confidential to Editor” section, and submit your "Accept" recommendation.

Reviewer #5: (No Response)

Reviewer #6: (No Response)

2. Is the manuscript technically sound, and do the data support the conclusions?

Reviewer #5: Partly

Reviewer #6: Yes

3. Has the statistical analysis been performed appropriately and rigorously? 

Reviewer #5: Yes

Reviewer #6: Yes

4. Have the authors made all data underlying the findings in their manuscript fully available?

Reviewer #5: Yes

Reviewer #6: Yes

5. Is the manuscript presented in an intelligible fashion and written in standard English?

Reviewer #5: Yes

Reviewer #6: Yes

6. Review Comments to the Author

Reviewer #5: PONE-D-19-30581_R1

The authors of “Impact of left ventricular assist devices and heart transplantation on acute myocardial infarction and heart failure mortality and readmission measures” sought to understand the frequency of deaths and readmissions in patients supported on advanced heart failure therapies as a component of Medicare population-based metrics to determine if inclusion of such patients unduly affects the Medicare-based assessment of those hospitals performance with respect to those measures. Certainly, these data are important to have available for consideration, but the way in which some of the data and conclusions are presented could be misleading and in the interest of disseminating this information, authors should seek to clarify this for potential authors who would cite these data.

Major Comments:

1. What was the authors’ intent with including temporary/external? LVADs? I would suggest that the authors completely exclude external or temporary LVADs (and why do the authors indicate in the limitations that they could not distinguish between percutaneous LVADs vs. durable VADs?). The authors ultimately argue that LVAD patients should be considered separately, but I am unsure if patients supported on temporary LVADs should be excluded given the controversy of using these devices (i.e. Impella) routinely. Protected PCI may be another indication during AMI admission for use of an “LVAD” which further complicates the data. Additionally, the controversy with respect to the HRRP primarily surrounds the HF admission indication and the observation of the disparate mortality and readmission data (as the authors also observed) perhaps unduly affecting high quality centers. I believe these data are of much more relevance wherein the proportion of durable LVADs is much higher.

2. However if the authors cannot readily separate temporary and durable device and so choose to proceed with inclusion of temporary LVADS, in the first methods paragraph, I would provide examples of the external or temporary LVAD versus semi-permanent or permanent LVADs for clarity for the reader to emphasize throughout the idea that markedly different LVAD devices with markedly different indications are lumped into his cohort.

3. Previous reviewers have criticized the manuscript for the statistical inability to generate differences between cohorts after removing LVAD and transplant patients. While I agree with the essence of these comments, I disagree about the resultant lack of importance of the manuscript. The manuscript, as I interpret it, is designed to provide a neutral assessment of the data in the CMS calculations. In that regard, the authors have done well in not applying a statistical test of significance. However, I would clarify that objective within the introduction and discussion. For example, rather than stating in the introduction that they “sought to examine the impact of …” I would suggest that the authors indicate that they sought to quantify the available data on LVAD and HTx mortality and readmissions as it relates to the CMS decision. Similarly, in the limitations, in the limitations, I would suggest that the authors indicate that due to very small numbers of patients in the cohorts of interest in a large administrative database, the study was not powered to detect differences and thus no statistical tests of significance were used.

Minor comment:

1. In the 3rd to last sentence of the first Methods paragraph, I believe the authors mean HTx cohort rather than HF cohort.

Reviewer #6: Brandt et al. have evaluated mortality and readmission measures of acute myocardial infarction and heart failure patients while assessing the impact of including patients receiving LVAD and heart transplants on these measures. The authors provide a revision of their manuscript along with responses to prior reviewer comments. The manuscript is well-written and presented in an intelligent fashion. The statistical methodology appears adequate although further description of the same in the methods section would be ideal. The authors have addressed most of the concerns raised in a reasonable manner. Most importantly it is apparent that they are limited by the data available to them with respect to providing granularity on this topic. Although I am inclined towards the author’s suggestion that these are important findings given their importance in policy making, I do have a few comments for the authors’ consideration,

Firstly, the choice of using data from 2010 to 2013 is surprising. The authors have addressed this concern in response to prior comments citing availability of data as an issue. However, as previously pointed out most devices part of this study are no longer in use and as such do not reflect current clinical practice. Given the authors primary objective was to provide basis for the CMS decision to exclude LVAD and heart transplant patients, I would urge them to include data up until 2016 as suggested earlier.

I am not entirely convinced with the conclusion that “Hospitals caring for >6 patients with LVAD or >1 HT typically had lower RSMRs but higher RSRRs” given the differences are less than 1% and the small cohort sizes (LVAD and heart transplant groups). Although the authors suggest that they are referring to this in a non-statistical manner, I would suggest saying “Hospitals caring for >6 patients with LVAD or >1 HT typically had a trend toward lower RSMRs but higher RSRRs”.

The authors acknowledge that their findings are expected and reiterate the importance of the findings to stake holders and decision makers. However, given the limitations of the database, the unavailability of data on types of LVAD, indications for use and the small number of patients receiving LVAD and heart transplant, it is hard to establish clinical precedence for this manuscript. Having said that, I would certainly urge the editor to consider this for publication provided the authors are able to add data up until 2016 when CMS made the policy change to provide more comprehensive findings representing that decision.

7. PLOS authors have the option to publish the peer review history of their article (what does this mean?). If published, this will include your full peer review and any attached files.

Reviewer #5: No

Reviewer #6: No

---

## [Author Response · Author response to Decision Letter 1]

5 Mar 2020

Additional Editor Comments (if provided):

1. The authors intended to assess the impact of LVAD and HTx, a subgroup, on some outcome measures. These surgical cases were a small minority and stat. insig changed those rates. In epidemiology, it is always better to enhance homogeneity to reduce selection bias. So it was not a bad idea to exclude those cases. But since their existence did not affect the rates significantly by statistical testing, why not just keep them to reflect the reality in clinical scenario? The authors needed to clarify the purpose of those outcome measures. Is there any benefit to exclude LVAD and HTx, such as for cost management, admin, insurance, etc? Please revise to reflect the motivation of the manuscript. LVAD and HTx, as a small minority, did not affect the outcome measures. Their non-exclusion did not violate homogeneity but also reflect the diversity of clinical reality. The authors need to provide a stronger reason to exclude them

Response: We appreciate the Editor’s questioning whether it was appropriate to exclude this small minority of patients, since inclusion in the models did not materially affect the hospital risk-standardized rates. The portion of the cohort that had LVAD and HTx were predictably small. However, our a priori methodology decision to exclude them and repeat the analysis was essential to addressing and being responsive to stakeholder and policymaker concerns, Providing reassurance that managing these patients does not lead to changes in hospital mortality and readmission since these measures could alter reimbursement and potentially patient care. We have made this argument more clear by addition to the introduction (page 5): “Exclusion of LVAD and heart transplant patients will inform whether these patients significantly alter hospital level mortality and readmissions, thus informing stakeholder and policymakers on the implications of having them included or excluded from future measures.”

Comments to the Author

Reviewer #5: PONE-D-19-30581_R1

The authors of “Impact of left ventricular assist devices and heart transplantation on acute myocardial infarction and heart failure mortality and readmission measures” sought to understand the frequency of deaths and readmissions in patients supported on advanced heart failure therapies as a component of Medicare population-based metrics to determine if inclusion of such patients unduly affects the Medicare-based assessment of those hospitals performance with respect to those measures. Certainly, these data are important to have available for consideration, but the way in which some of the data and conclusions are presented could be misleading and in the interest of disseminating this information, authors should seek to clarify this for potential authors who would cite these data.

Major Comments:

1. What was the authors’ intent with including temporary/external LVADs? I would suggest that the authors completely exclude external or temporary LVADs (and why do the authors indicate in the limitations that they could not distinguish between percutaneous LVADs vs. durable VADs?). The authors ultimately argue that LVAD patients should be considered separately, but I am unsure if patients supported on temporary LVADs should be excluded given the controversy of using these devices (i.e. Impella) routinely. Protected PCI may be another indication during AMI admission for use of an “LVAD” which further complicates the data. Additionally, the controversy with respect to the HRRP primarily surrounds the HF admission indication and the observation of the disparate mortality and readmission data (as the authors also observed) perhaps unduly affecting high quality centers. I believe these data are of much more relevance wherein the proportion of durable LVADs is much higher.

Response: We thank the reviewer for the thoughtful discourse about our work. The reviewer recognizes the importance of our manuscript, that the populations for which LVADs are used have different indications for use as well as different clinical scenarios. This is precisely why our data were important for CMS and why temporary/external LVADs were important to include. We found that LVADs used in the AMI cohort more often had indications for temporary rather than chronic support needs. The data do not inform the exact type of LVAD nor whether the devices were semi-permanent or permanent. This limitation of the data cannot be overcome, nor are there additional data available that would refine our ability to differentiate the exact device type. Excluding temporary/external LVADs from the data would significantly alter the relevance of our findings, a portion of which were to inform on decisions to include or exclude LVAD patients from the AMI cohort. The majority of the devices in the HF cohort were implanted device. Even when including patients with external/temporary LVADs, which would bias us away from our current conclusions, the data still support excluding these patients. Thus, we expect our findings are more robust by continuing to include these patients. 

2. However if the authors cannot readily separate temporary and durable device and so choose to proceed with inclusion of temporary LVADS, in the first methods paragraph, I would provide examples of the external or temporary LVAD versus semi-permanent or permanent LVADs for clarity for the reader to emphasize throughout the idea that markedly different LVAD devices with markedly different indications are lumped into his cohort.

Response: As discussed above, we cannot readily separate these devices. We have edited the text to provide examples of devices available at the time of the study. Page 4: “Some devices are delivered percutaneously for temporary hemodynamic support (i.e. Impella® and TandemHeart® devices at the time of the study), while others are implanted for intermediate to longer term support as destination therapy or bridge to transplant (i.e. HeartMate® II, at the time of our study).”

3. Previous reviewers have criticized the manuscript for the statistical inability to generate differences between cohorts after removing LVAD and transplant patients. While I agree with the essence of these comments, I disagree about the resultant lack of importance of the manuscript. The manuscript, as I interpret it, is designed to provide a neutral assessment of the data in the CMS calculations. In that regard, the authors have done well in not applying a statistical test of significance. However, I would clarify that objective within the introduction and discussion. For example, rather than stating in the introduction that they “sought to examine the impact of …” I would suggest that the authors indicate that they sought to quantify the available data on LVAD and HTx mortality and readmissions as it relates to the CMS decision. Similarly, in the limitations, in the limitations, I would suggest that the authors indicate that due to very small numbers of patients in the cohorts of interest in a large administrative database, the study was not powered to detect differences and thus no statistical tests of significance were used.

Response: We appreciate the Reviewer’s comment and recognize that we had not made this point clear in our prior version of our manuscript. We have edited the manuscript with the language suggested (page 7): “Due to the small numbers of patients with LVAD and heart transplantation in the cohorts, the study was not powered to detect differences and thus no statistical tests of significance were used.”

Minor comment:

4. In the 3rd to last sentence of the first Methods paragraph, I believe the authors mean HTx cohort rather than HF cohort.

Response: The reviewer is correct and we had made an error in this sentence. We have corrected this error, see page 5: “Inclusion in the cohorts was defined by ICD-9CM procedure codes 37.60, 37.62, 37.65. 37.66. and 37.68 for the LVAD cohort and 33.6 and 37.51 in the heart transplant cohort.”

Reviewer #6: Brandt et al. have evaluated mortality and readmission measures of acute myocardial infarction and heart failure patients while assessing the impact of including patients receiving LVAD and heart transplants on these measures. The authors provide a revision of their manuscript along with responses to prior reviewer comments. The manuscript is well-written and presented in an intelligent fashion. The statistical methodology appears adequate although further description of the same in the methods section would be ideal. The authors have addressed most of the concerns raised in a reasonable manner. Most importantly it is apparent that they are limited by the data available to them with respect to providing granularity on this topic. Although I am inclined towards the author’s suggestion that these are important findings given their importance in policy making, I do have a few comments for the authors’ consideration,

1. Firstly, the choice of using data from 2010 to 2013 is surprising. The authors have addressed this concern in response to prior comments citing availability of data as an issue. However, as previously pointed out most devices part of this study are no longer in use and as such do not reflect current clinical practice. Given the authors primary objective was to provide basis for the CMS decision to exclude LVAD and heart transplant patients, I would urge them to include data up until 2016 as suggested earlier.

Response: We agree that using the most available data is always best, when possible. The reviewer points out that the devices used today differ from that during our study period. However, the frequency of device use has also changed since the period from 2014 through 2016. Thus, we don’t think that adding data through 2016 would reflect on current device utilization either since the most frequently utilized implanted LVAD (the Heartmate III) was not approved by the FDA for use until 2017. Lastly, the decision to exclude LVAD and heart transplant patients was based on the data presented in our manuscript and not data from after 2013. Thus, we feel it makes the most sense to present on the data that were utilized in this decision. Finally, but also critically, data from after 2013 are not currently available to the authors. Obtaining agreement to analyze post-2013 would require additional contracting and is not feasible. 

2. I am not entirely convinced with the conclusion that “Hospitals caring for >6 patients with LVAD or >1 HT typically had lower RSMRs but higher RSRRs” given the differences are less than 1% and the small cohort sizes (LVAD and heart transplant groups). Although the authors suggest that they are referring to this in a non-statistical manner, I would suggest saying “Hospitals caring for >6 patients with LVAD or >1 HT typically had a trend toward lower RSMRs but higher RSRRs”.

Response: We agree with the reviewer that the altered statement more accurately reflects the data. Thus, we have updated the manuscript accordingly in both the abstract and the conclusion. See pages 3 and 19, which now read as follows: “Hospitals caring for >6 patients with LVAD or >1 patients with heart transplantation typically had a trend toward lower RSMR but higher RSRR.” and “…hospitals caring for patients with LVAD or performed heart transplants typically had a trend toward lower RSMR but higher RSRR,” respectively.

3. The authors acknowledge that their findings are expected and reiterate the importance of the findings to stakeholders and decision makers. However, given the limitations of the database, the unavailability of data on types of LVAD, indications for use and the small number of patients receiving LVAD and heart transplant, it is hard to establish clinical precedence for this manuscript. Having said that, I would certainly urge the editor to consider this for publication provided the authors are able to add data up until 2016 when CMS made the policy change to provide more comprehensive findings representing that decision.

Response: We thank the reviewer for recognizing the importance of our findings for stakeholders and decision makers. Please see our complete response above as to why it is not feasible to include data through 2016.

---

## [Decision Letter · Decision Letter 2]

9 Mar 2020

Impact of left ventricular assist devices and heart transplant on acute myocardial infarction and heart failure mortality and readmission measures

PONE-D-19-30581R2

Dear Dr. Desai,

We are pleased to inform you that your manuscript has been judged scientifically suitable for publication and will be formally accepted for publication once it complies with all outstanding technical requirements.

With kind regards,

Saraschandra Vallabhajosyula, MD FACP

Academic Editor

PLOS ONE

Additional Editor Comments (optional):

Reviewers' comments:

Reviewer's Responses to Questions

**Comments to the Author**

1. If the authors have adequately addressed your comments raised in a previous round of review and you feel that this manuscript is now acceptable for publication, you may indicate that here to bypass the “Comments to the Author” section, enter your conflict of interest statement in the “Confidential to Editor” section, and submit your "Accept" recommendation.

Reviewer #5: All comments have been addressed

Reviewer #6: All comments have been addressed

2. Is the manuscript technically sound, and do the data support the conclusions?

Reviewer #5: Yes

Reviewer #6: Yes

3. Has the statistical analysis been performed appropriately and rigorously? 

Reviewer #5: Yes

Reviewer #6: Yes

4. Have the authors made all data underlying the findings in their manuscript fully available?

Reviewer #5: Yes

Reviewer #6: Yes

5. Is the manuscript presented in an intelligible fashion and written in standard English?

Reviewer #5: Yes

Reviewer #6: Yes

6. Review Comments to the Author

Reviewer #5: The authors have addressed all concerns of this reviewer. The authors appropriately recognize the limitations of their dataset in the manuscript and the conclusions are not overstated. The manuscript continues to be well-written.

Reviewer #6: (No Response)

7. PLOS authors have the option to publish the peer review history of their article (what does this mean?). If published, this will include your full peer review and any attached files.

Reviewer #5: No

Reviewer #6: No

---

## [Editor Report · Acceptance letter]

12 Mar 2020

PONE-D-19-30581R2 

Impact of left ventricular assist devices and heart transplants on acute myocardial infarction and heart failure mortality and readmission measures 

Dear Dr. Desai:

I am pleased to inform you that your manuscript has been deemed suitable for publication in PLOS ONE. Congratulations! Your manuscript is now with our production department. 

With kind regards,

on behalf of

Dr. Saraschandra Vallabhajosyula 

Academic Editor

PLOS ONE